# The HEADS: UP Development Study: Working with Key Stakeholders to Adapt a Mindfulness-Based Stress Reduction Course for People with Anxiety and Depression after Stroke

**DOI:** 10.3390/healthcare11030355

**Published:** 2023-01-26

**Authors:** Maggie Lawrence, Bridget Davis, Leyla De Amicis, Jo Booth, Sylvia Dickson, Nadine Dougall, Madeleine Grealy, Bhautesh Jani, Margaret Maxwell, Ben Parkinson, Matilde Pieri, Stewart Mercer

**Affiliations:** 1Research Centre for Health (ReaCH), Glasgow Caledonian University (GCU), Glasgow G4 0BA, UK; 2School of Education, University of Glasgow, Glasgow G3 6NH, UK; 3Health and Social Care Sciences, Edinburgh Napier University, Edinburgh EH11 4BN, UK; 4Psychological Services and Health, University of Strathclyde, Glasgow G1 1XQ, UK; 5General Practice and Primary Care, School of Health and Wellbeing, MVLS, University of Glasgow, Glasgow G12 9LJ, UK; 6Nursing, Midwifery and Allied Health Professions Research Unit (NMAHP-RU), Faculty of Health Sciences and Sport, University of Stirling, Stirling FK9 4LA, UK; 7Usher Institute, University of Edinburgh, Edinburgh EH8 9AG, UK

**Keywords:** stroke, mindfulness-based stress reduction, anxiety, depression, co-development, adaptation, complex intervention

## Abstract

Background: Following stroke, rates of mood disorder are and remain high at five years (anxiety 34.4%; depression 23%). Structured mindfulness-based stress reduction (MBSR) courses are effective in a range of health conditions, but stroke survivors find adherence challenging. We aimed to adapt a standard MBSR course specifically for people affected by stroke. Methods: We recruited stroke survivors and family members with symptoms of anxiety and/or depression to take part in a co-development study comprising two rounds of MBSR ‘taster’ sessions, followed by focus groups in which views were sought on the practices sampled. Data were collected in October 2017 and May 2018 and were analysed using framework analysis, informed adaptations to mindfulness materials and delivery. Results: Twenty-eight stroke survivors and seven family members participated. Nineteen (76%) stroke survivors had anxiety; 15 (60%) had depression. Five (71.4%) family members reported anxiety; *n* = 4 (57.1%) depression. Thirty participants attended the first round of taster sessions and focus groups; twenty (66%) the second and three (10%) were unable to attend either round. Framework analysis informed adaptations to course delivery, practices, and materials, ultimately resulting in a stroke-specific MBSR course, HEADS: UP (Helping Ease Anxiety and Depression after Stroke). Conclusions: HEADS: UP may provide a feasible, appropriate, and meaningful self-management intervention to help alleviate symptoms of mood disorder.

## 1. Introduction

Stroke is a leading cause of death and disability [1]. Globally, over 80 million people live with the effects of stroke and annually new events affect 13.7 million people [1]. One in four people aged over 25 years will have a stroke in their lifetime [2]. The potential effects are many and varied, and may include physical, cognitive, and communication impairment as well as psychological disorders of anxiety, depression and psychosocial stress, the last often being referred to as ‘invisible’ effects of stroke [3]. An additional feature of stroke is recurrence; in a UK population-based cohort study (*n* = 6052), estimated 12% (10%–15%) at 5 years [4]. It is estimated that up to 90% of stroke events are linked to lifestyle risk factors, including psychosocial factors (stress (home and work), life events, and depression), and could be preventable if people took action to reduce risk such as making changes to lifestyle factors including stress management [5].

At a systems level it is estimated that per patient month stroke services and care provision costs USD 4850 (USA), or AUD 752 (Australia) [6]. At the individual level psychosocial wellbeing can be significantly challenged, with individuals experiencing psychological disorders, distress, and social isolation, which individually or in combination may affect long-term functioning and quality of life [7]. Stroke impacts are felt across families, friends, and wider networks who may experience a sense of burden, emotional distress and disruption of family relationships [8,9]. Stroke experiences are different for everyone, and recovery trajectories are variable [10]. Psychological consequences of stroke are considerable and often enduring. A large-scale meta-analysis (22,262 participants; 34 countries) found post-stroke anxiety in 18.7% (95% CI, 12.5, 24.9) with clinical interview and 24.2% (95% CI, 21.5, 26.9) using a rating scale [11]. A 5-year post-stroke meta-analysis revealed depression in 23% (95% CI, 14%–31%) of cases [12]. A large cohort study [13] found anxiety at five years in 34.4% (95% CI, 30.8%–38.1%). Mood disorder negatively impacts rehabilitation outcomes, possibly due to reduced engagement levels [14,15] and on proactive self-management of recovery and prevention of recurrence of stroke and other cardiovascular disorders [15,16]. A recent systematic review identified that low mood/ and depression negatively affected motivation to be active, even though being physically active is known to improve mood [17]. In a stakeholder consultation project, stroke nurses identified management of mood as a research priority [18], and people affected by stroke and health professionals, identified long-term psychological consequences of stroke as an unmet need and called for long-term, family-centred, self-management interventions [19]. 

Mindfulness-based interventions (MBI) are structured mindfulness courses, developed from traditional Buddhist meditation [20]. The first mindfulness-based stress reduction (MBSR) course was developed in the 1980s [21]. Forty years on, MBSR is a widely used intervention delivered to groups in clinical and non-clinical settings. It has an established protocol [22] comprising an introductory session followed by eight weekly sessions (2.5–3.5 h; 7.5 h (day retreat) in week 6). Daily practice, sustained over time, is an essential element of course engagement. A considerable body of evidence demonstrates the effectiveness of MBSR, and more broadly, MBIs in the treatment of mood disorder (predominantly anxiety and depression) in diverse clinical populations, including in long-term conditions ((95% CI, 0.35, 0.17–0.53) [23], and in a range of settings from acute clinical settings to the wider community [24].

In relation to stroke, research evidence regarding psychosocial complex interventions is inconclusive and often based on methodologically weak studies [25]. Lawrence et al.’s review (2013) found only four studies, three of which were methodologically poor [26]. No statistically significant findings were reported; however, positive trends for psychosocial complex interventions were observed across a range of outcomes including anxiety and depression. Study attrition rates were high (23%–61%) and intervention adherence was poor, often due to common stroke-related issues including fatigue, cognitive problems, and travel difficulties—challenges reported in other studies (e.g., [17,27]). While arguably, MBSR is not a self-management intervention per se, it does incorporate principles of self-management including self-efficacy [16]. The course facilitates learning new skills which, with practice, can lead to mastery of those skills that can be used by individuals to self-manage symptoms of mood disorder in the longer term. 

Cognizant of the size and nature of the problem, the context of the increasing relevance of (supported) self-management of chronic conditions [28,29], and substantial evidence highlighting the potential for MBSR to be an effective, appropriate, and meaningful intervention for people affected by stroke [30,31], a programme of work, HEADS: UP (Helping Ease Anxiety and Depression after Stroke), was conceived. The overarching aim of the programme is to ascertain the effectiveness and appropriateness of HEADS: UP, a stroke-specific adaptation of a standard MBSR course. This paper reports the processes and outcomes of the development stage in which we aimed to adapt, or tailor, a standard MBSR course to enhance its appropriateness and meaningfulness for people affected by stroke. 

## 2. Materials and Methods

In this development work we drew on the Medical Research Council’s guidance for complex intervention design [32,33]. Accordingly, in earlier preparatory work we had reviewed the stroke/MBSR evidence base [26] and identified conceptual and theoretical understandings which would inform aspects of the work, namely family systems theory [34], behaviour change theory [35,36] and self-management [37,38]. This preparatory work formed the underpinning to the empirical development study reported here. It should be noted that we use the term ‘adaptation’ to describe the work rather than ‘development’, as we were working to enhance the accessibility, acceptability, and meaningfulness of a pre-existing standardised intervention [39]. However, the work is dynamic, iterative, creative, open to change, and is forward looking to future evaluation and implementation, and as such meets development criteria described by O’Cathain and colleagues [33]. Protocol Registration: ClinicalTrials.gov Identifier NCT04985838.

### 2.1. Aim, Design and Setting

In this community-based study we aimed to adapt a standard MBSR course to enhance its appropriateness and meaningfulness for people affected by stroke. We hypothesized that: (i) adapting the standard MBSR course to enhance its relevance to people affected by stroke; (ii) incorporating a supportive mechanism, i.e., working with a partner; and (iii) integrating behaviour change techniques would help improve stroke survivors’ engagement with, and adherence to, an MBSR course, course materials, and the requirement for daily personal practice. 

We used qualitative and co-creation methods in an iterative design (see Figure 1), in which we combined three intervention development approaches, namely target population centered; theory and evidence based; and implementation based [33].

In an iterative process involving two rounds of ‘taster sessions’ of MBSR (see Figure 1) we exposed stroke survivors and family members to examples, or ‘tasters’, of core elements that comprise an MBSR course. The practices, e.g., body scan, mindful movement, breath meditations, and seated and walking meditations represent the core practices of secular mindfulness [22]. Prior to attendance at an MBSR taster session, a study-specific ‘Introduction to Mindfulness’ information sheet was sent to participants, along with accessible directions to the venue (Appendix A). Taster sessions lasted 75 min (approx.) and were led by qualified MBSR trainers. Equipment (yoga mats, lightweight blankets, foam blocks) was provided to support practice, if required. ML (first author) accompanied the trainer during the sessions to provide support should participants have needed assistance with, for example, mobility. Following the taster sessions, participants were provided with refreshments and then invited to take part in focus groups (see below for more detail). Taster sessions and focus groups were delivered in accessible community venues in Central and West Scotland, with good transport links. Taxis were provided, if requested.

Following analysis of focus group data and consultation with a team of experts representing diverse range of stakeholders (i.e., lay representatives, representatives of third sector organisations, clinicians, and MBSR trainers [25]), we made changes to taster session materials and delivery, incorporating behaviour change techniques as appropriate [38]. Once the iterative cycle of taster sessions and adaptation was complete, we extrapolated from these evidence-based and theoretically informed amendments and applied the changes across the whole course (materials and delivery) to create the stroke-specific MBSR course, HEADS: UP (Helping Ease Anxiety and Depression after Stroke) [40,41].

### 2.2. Recruitment

We used a community-based recruitment strategy, principally comprising third sector organisations (stroke and carers’ organisations) supplemented by social media, to recruit adult stroke survivors aged ≥ 18 or adult family members of stroke survivors. The strategy allowed for additional snowballing [42]. Gatekeepers distributed a letter outlining the aim and purpose of the study and explaining how to contact the research team. An abbreviated invitation was used for social media routes. Individuals expressing an interest in participation were contacted by the researchers who either posted out a participant information leaflet and consent form, or arranged to visit the individual, according to their preference. Aphasia-friendly paperwork was used, as appropriate. The research team discussed the study with the potential participants, answered any questions, and if appropriate collected informed consent. 

Participants were required to have self-reported and/or clinically diagnosed stroke and self-reported and/or clinically diagnosed mood disorder and an interest in/or experience of MBSR. We included stroke survivors with mild-to-moderate communication impairments but excluded potential participants if they had severe communication impairment(s) and/or were currently accessing any rehabilitation service. 

We aimed to recruit stroke survivors and family members as either dyads or as individuals participating in the study without an accompanying ‘other’. Stroke survivors who expressed interest in the study were invited to recruit a family member, if desired. Likewise, family members who expressed interest in the study were invited to recruit a relative who had had a stroke, if they wished. Family members were defined as “*‘a self-identified group of two or more individuals who consider themselves as family’, e.g., a spouse, parent, friend or neighbour.*” [43]. Family members were recruited to the study in their own right, i.e., eligibility did not rely on having a family member who had had stroke also participating in the study. 

### 2.3. Data Collection 

Informed consent was obtained ahead of collection of demographic data (gender, age, time post-stroke, symptoms of anxiety (yes/no), symptoms of depression (yes/no)) and participation in taster sessions and focus groups. Focus groups have been used previously to engage people affected by stroke in co-creation work, including intervention development [44]. For this study, a semi-structured focus group schedule (Appendix A) was used to guide and facilitate discussion between participants [45,46]. We aimed to explore facilitators and challenges associated with mindfulness practice and adherence to/development of personal practice habits or behaviours. In terms of facilitators, we aimed in particular to explore perceptions regarding dyadic attendance, and whether this was considered likely to enhance stroke survivors’ adherence to the course and to personal practice, and whether the family members themselves might derive therapeutic benefit from attending the course as part of a dyad. Focus group discussions, moderated by BD and co-moderated by LDA (second and third authors) were audio recorded and then transcribed. The role of the moderator (BD) was to ‘manage’ the focus group, e.g., ensure all participants had the opportunity to contribute to the discussions. The co-moderator (LDA) supported BD, e.g., by making field notes and asking any supplementary questions, if appropriate; neither were involved in the taster sessions.

### 2.4. Iterative Process of Data Analysis and Adaptation of Materials

Thinking ahead to future implementation work [33], we used framework analysis methods [47] to analyse the focus group data. Our framework was based on selected domains of TIDieR (Template for Intervention Description and Replication) [48], i.e., What? (materials and procedures), Who? (provider), How? (mode of delivery), Where? (location), and When? (intensity and duration), and Tailoring (modifications made during delivery/fidelity). TIDieR domains were supplemented by inductive themes developed during the iterative process of analysis, providing a broader understanding of the participant experience which included exploration of dyadic attendance as a supportive mechanism. In the interests of rigor, two researchers (BD and ML) independently immersed themselves in the data by reading the transcripts and listening to the audio files before independently coding the data using the TIDieR/inductive framework. Any differences in coding were subsequently resolved through discussion. This process enabled us to identify data which would inform adaptations to course materials and delivery to enhance accessibility for stroke survivors. 

### 2.5. Adaptation of the Intervention

Generic adaptations reflecting contemporaneous accessible information guidance [49,50], including accessible communication recommendations from the stroke-related aphasia literature [51,52], informed adaptations to the manual content, layout, and formatting, e.g., using white space, large font, left justification of text, short sentences and plain English. Aphasia accessibility guidelines informed the use of images to illustrate content, and emboldened text to draw attention to key words and concepts. An additional layer of adaptation was informed by our analysis and interpretation of the qualitative data, which formed the basis of subsequent adaptations. Using the results of our mapping of the behaviour change techniques (BCTs) implicit and explicit in the original manual to highlight potential ‘gaps’, we incorporated additional BCTs to support behaviour change—in this instance development of a mindfulness ‘habit’.

To manage and report these iterative rounds of adaptation we developed an intervention modelling matrix (IMM) which brought together key aspects of the adaptation process. We used the IMM to record materials identified for adaptation along with the corresponding supportive qualitative data, and potential solutions/amendments identified in consultation with a team of experts. We then recorded participants’ responses to these amendments and any changes made subsequently. We worked to incorporate behaviour change techniques and mechanisms, and other accessibility features that could help people follow the course and embed mindfulness practice into daily life. An excerpt of the completed IMM is presented in the Results below. The team of experts comprised a diverse range of stakeholders (i.e., lay representatives, representatives of third sector organisations, clinicians, and MBSR trainers). The individual experts were network contacts or were recommended by network a contact.

Thinking ahead to future evaluation and implementation, a HEADS: UP Train the Trainer training package (not yet reported) was developed, informed by this adaptation work. The prototype training package aimed to inform experienced MBSR trainers about the background to the HEADS: UP programme of research and relevant research processes. Importantly, it also aimed to raise awareness of potential impacts and effects of stroke, and how these might affect course participants and their interaction(s) with the trainers, and hence the trainers’ considerations about how the course is delivered. 

## 3. Results

Recruitment occurred over 6 weeks in August and September 2017. We received 38 expressions of interest. Three were lost to contact; 35 (92.1%) gave written informed consent to participate in the study. Of these, 2 (5.7%) were unable to provide data/take part due to ill health/family reasons and 1 (2.9%) family member was lost to contact. Thirty-two (91.4%) participants provided demographic data and took part in ≥1 taster session/focus group; 25 (78.1%) were stroke survivors and 7 (21.9%) were family members. Stroke survivor participants were female (16, 64%), aged 62.9 years (SD 7.7), and 46 months post-stroke (median, IQR 12.5–129.5 months). Nineteen (76%) reported having anxiety; 15 (60%) depression. Family members were female (5, 71.4%) and aged 63.2 years (SD 19.6). Five (71.4%) reported having anxiety; 4 (57.1%) had depression (see Table 1 and Table 2 for full participant demographic data).

In October 2017, and again in April 2018, we held three taster sessions and focus groups, one in a community venue and two on the university campus. Group size varied according to geographic location and participant availability on the day, i.e., (October/April) Group 1 *n* = 10/5, Group 2 *n* = 12/9, Group 3 *n* = 8/6. Thirty (85.6%) participants attended the first round of taster sessions and focus groups; *n* = 1 had died; *n* = 2 cancelled due to ill-health; *n* = 2 did not attend/not available. Twenty (66%; stroke survivors *n* = 16; family members *n* = 4) participated in the second round; 18 (56%) who had taken part in October, along with two participants unable to attend at that time.

### 3.1. Qualitative Findings

Focus group data were analysed using as the deductive framework the six domains of the TIDieR listed above, which were supplemented by three inductive themes identified during the analysis process, *The role of Mindfulness in mitigating stroke risk factors and effects, Engaging with Mindfulness, Taking part with an Other*. The inductive themes only are reported here, supported by verbatim quotes. Information in brackets details whether the quote comes from the October 2017 (Oct) or April 2018 (Apr) focus group, the ‘status’ (stroke survivor (SS); family member (FM)), participant identity number assigned during recruitment, and gender of the participant.

#### 3.1.1. The Role of Mindfulness in Mitigating Stroke Risk Factors and Effects

The potential for mindfulness practice to impact positively on challenging behaviours, attitudes, and habits was discussed. Participants thought that practicing techniques learnt in the taster session and learning more about mindfulness in the future could have a range of benefits including helping them better manage anger and stress, and even help prevent recurrent stroke. One ‘angry’ family member who attend the taster session with his father, a stroke survivor, explained that being easily roused to anger was something he and his father had in common. He hoped that mindfulness would help them both address their anger, and ultimately prevent the occurrence of further strokes, a sentiment echoed by his father:


*The reason why I really wanted to come here, because me and my dad share the same sort of character traits. We both tend to get quite angry, pretty quickly, about things that don’t really matter … So, for me, I’m interested in this [mindfulness] because I don’t want to, you know, perpetuate [this behaviour], as I get older, and I think that might be one of the reasons why you [addressing his father] could have been stressed and had your stroke. So, yeah, the sitting down, and sort of focusing on something, and being able to completely relax, I found really beneficial. (Oct, FM36, male) …*



*I would agree with that. That is not where I want to be. Being in that state of mind I become judgmental. There’s ‘my’ road map of how the world should look, and when people, and things, don’t conform to that, it annoys me. Why? Stop judging! And I think if you take the time every day, to concentrate on yourself, you can probably get into that (relaxed) state of mind more easily.” (Oct, SS33, male)*


Participants discussed perceived psychosocial stress experienced within their busy lives. Some considered the mindfulness techniques taught in the taster session might be helpful in relieving stress in the short-term by helping them to ‘still’ their cluttered minds. They also considered the possibility of developing a more long-term mindfulness practice and the potential for this to help better manage feeling of stress in the future, as discussed by these two participants:


*I find it very hard to relax and switch my mind off everything. My mind is very busy at all times. Like [in the taster session] at first I was thinking, what I was gonna do today after this, what shopping I needed, what I was gonna do. And then, I just brought my mind back to, you know, just thinking about, concentrating on your breathing … and everything. And I found that really good. So, I can do this (Oct, SS14 female) …*



*… Yes, I think you can! I think it takes a lot of effort, but … I think everybody’s life is quite stressed, and I think if you can master mindfulness you’re going to have a much less stressful life and that’s what I felt was how I want it to change my life (Oct, SS25 female)*


Participants also considered that mindfulness practice might help mitigate some effects of stroke such as cognitive impairment and fatigue. Cognitive impairment was discussed in terms of difficulties experienced with concentration, which participants found impacted negatively on their everyday lives. One participant described the impact in the context of return to work and speculated that practicing mindfulness could help improve concentration. 


*[since my stroke] I find that after a few minutes my mind is all over the place and I’ve lost my place in the paper that I’m reading. And so, I would hope that I would develop from this [taster session] that sort of ability to focus more clearly on what I do. I think [Mindfulness] is something we should be trying to do … because loss of concentration and focus has been a big thing in my ability to go back to work (Oct, SS4, male)*


However, some participants, including those with mild aphasia, found that time spent trying to concentrate often led to fatigue and therefore, the concentration required to practice mindfulness might actually cause fatigue. The group agreed and suggested that to address this mindfulness sessions should incorporate breaks:


*I’m aphasic … and when I’m fatigued, you know … it takes longer for me [to take things in] (Oct, SS19 female)*



*And if you’re concentrating really hard, I mean, it can make you feel more tired than what you usually feel (Oct, SS14 female)*



*But it [a long session] might be ok, if you get out for five, ten minutes outside, to get a wee bit of air (Oct, SS16 female)*


#### 3.1.2. Engaging with Mindfulness

Participants described how the pre-session information sheet had allowed them to focus on mindfulness and prepare for the taster session; however, others felt it was too brief and they had not felt sufficiently informed about what taking part in the session might entail.

Following revision undertaken in between the iterative cycles of taster sessions/focus groups, the ‘HEADS: UP Introduction to Mindfulness’ information leaflet which included detail about ‘What is mindfulness?’ was well received. Participants found the content clear and volume of information sufficient for them to feel adequately informed:


*It gave me all the information I needed. I felt as prepared as possible. I didn’t need any more information (Apr, SS25, female)*



*it gave us more information, without the jargon, without lots of jargon, yeah (Apr, FM18, female)*


Following the first round of taster sessions, the pre-session information was further developed to include a short list of mindfulness resources, which included books and links to reputable websites:


*What I found most helpful out of all the information you sent was the links to the websites, so I used the Mindfulness in Scotland [sic] one. I used that in between the two [taster] sessions, and I found that very helpful rather than say … reading a book (Apr, SS11, female)*


Reflections about the actual mindfulness taster sessions revealed a range of experiences and responses across participants, from general enjoyment of the practices to uncertainty about whether mindfulness was something they wanted to pursue. For example, some felt that one taster session was not enough for them to make a decision one way or another, and one participant expressed a preference for her own familiar coping strategies:


*But it’s not personally for me … I have my own coping strategies that I do at home (Oct, FM32, female)*


Participants’ reactions to specific practices varied; often this was related to effects of stroke that negatively impacted their ability to engage with some of practices. ‘Mindful walking’ was particularly divisive:


*what I’m gonna do is try and plug [mindful walking] into my day. And every so often, just get up, and just walk about, and use mindfulness walking, because that will also help me with the concentration (Apr, SS4, male)*



*I find … I’ve always got to concentrate whenever I’m walking, what foot I need to use, you know, to go up or go down, or whatever. I remember someone saying that to me, once, ’Do you always have to think about that?’, and I said,’ I do’, and I still do, have to think about it … so I just sat there [during mindful walking practice] (Apr, SS3, female)*


In response to this feedback, changes were made to the delivery of the practice, such as providing a clear explanation of what was to come and clearly describing and demonstrating alternative ways of engaging with the practice. 


*The first session we had was a wee bit … like just go straight on in there, but this time … there was more explanation … “This is what we’re thinking about, this is hopefully what you’re going to get from it, the theory of that is …” … so from that point of view, I found it better (Apr, SS1, male)*


Initially, participants found that they needed more time to take in and absorb instructions given before a practice. In the second round of taster sessions delivery was slower, instructions were repeated, and advice was given about different ways of engaging with a practice. Most participants found this change helpful: 


*And I think, just the way he explained, really thoroughly, he explained what he was doing, and advising us how to go about things … it was, you know, really good, helpful (Apr, SS27, female)*



*I found this much better … I was just being talked through different stages (Apr, SS38, male)*


Although not everyone agreed:


*I found the other [mindfulness exercises] difficult … because I didn’t understand the information. Because I’ve never done it before, and I didn’t know what I was doing (Apr, SS34, female)*


Practicing mindfulness skills learned on the course in your own time is essential, as it is with acquiring any new skill. Participants reported struggling to prioritize personal practice within the demands of their busy lives: 


*I can’t really do it at home, I try, but I fail abysmally, I just can’t do it somehow. But I need to … I know. Whether it’s because I’m at home, you know, and thinking, I should be doing this, or I could go and do that. I need to forget these things, but it’s difficult (Apr, SS17, female)*


Memory impairment was mentioned frequently and particularly in relation to personal practice. Participants suggested that having reminders, including text prompts, might be a way to address this:


*Yeah, well again, it was remembering to do it … I think it would be useful to get a reminder … to say, “Get in and do your mindfulness!” you know, it would be good … [signs of agreement from SS29, SS34, SS4] … that would be good (Apr, SS4, male)*


Another pertinent and related issue discussed by participants was the practicalities of how mindfulness practice could be incorporated into everyday life:


*Just how do you actually, you know, what do you do on a day-to-day basis? Are you meant to take ten minutes in the morning and do that, or when you’re sitting in your car and someone’s annoyed you, do you pull over to the side of the road and do it, you know? How does it sort of practically merge into your life …?" (Oct, FM36, male) … In response one participant suggested, You could do it on a bus, you could do it anywhere, couldn’t you? (Oct, SS22, female)*


#### 3.1.3. Taking Part with an Other

Some participants attended the taster sessions as dyads, i.e., with a family member or friend. Some dyadic stroke survivor participants were concerned about the burden this level of commitment would place on their family member or carer:


*I think it’s quite a time commitment to ask [of] somebody. It’s all right for me to do it, but to ask somebody else, like a friend or a family member to do it, it’s quite a time commitment for them (Apr, SS10, female)*


Conversely, family member or carers were more likely to be concerned with supporting their relative giving little or no thought to the benefit they themselves might derive from engaging with the mindfulness practices. However, one family member who attended with his wife (a stroke survivor) did consider that mindfulness could be of benefit to them both and described how, by attending together they could provide mutual support for one another:


*[I need to concentrate on] my wife, her needs, my own needs, and how we can best do it for each other (Oct, FM26, male)*


Participants considered whether experiencing mindfulness with an ‘other’ unrelated to them would be possible, and a ‘within group’ or ‘buddy’ approach was discussed:


*Not everybody could have family and friends available [to attend with], maybe they [the ‘other’] don’t want to get involved … maybe we can go into wee teams, so maybe there are three of us, we’ll sit together … rather than bring somebody (Apr, SS1, male) … I think the idea of maybe having a system within the group, where not everybody has to bring somebody with them, but you can support each other would work quite well (Apr, SS10, female)*


However, participants also identified drawbacks to attending as part of a dyad, in particular the need for privacy and to avoid unwanted disclosure.


*You may have a very good friend and family member, but there may be something that you actually didn’t want to discuss with them. So, there’s a wee danger there (Apr, SS1, male)*


### 3.2. Adapting the Intervention

Analysis of the qualitive data informed amendments made to the intervention at two different stages of the research: firstly, after the first round of taster sessions and focus groups, and again after the second round of taster sessions and focus groups in which participants experienced and then discussed the amendments made to the delivery of the practices and to the supporting materials. Table 3 below provides an illustrative example of how the qualitative findings informed changes to the course and supporting materials, and to Train the Trainer training materials.

## 4. Discussion

Using an iterative process, this study of intervention adaptation resulted in co-production of a new mindfulness-based intervention, HEADS: UP (Helping Ease Anxiety and Depression After Stoke). HEADS: UP is an adapted version of MBSR for people affected by stroke that aims to support engagement with, and adherence to, the course and development of mindfulness skills to help reduce symptoms of anxiety and depression. We worked collaboratively with people affected by stroke and other experts to enhance the appropriateness and meaningfulness of the adaptations and to explore aspects of feasibility that would inform design of the next stage of research [25,53]. 

In addition to the physical effects of stroke, stroke survivors described common consequences of stroke such as cognitive impairment, fatigue, and aphasia, and suggested changes which when applied to a standard MBSR course could enhance its appropriateness and meaningfulness. Cognitive impairment can cause stroke survivors to feel overwhelmed by information volume and speed of delivery. Participants in this study suggested amendments that would lighten the cognitive load and improve accessibility more generally. These included shorter sessions, a slower pace of delivery that is more stroke-aware and which incorporates a greater degree of repetition, using simplified language and less jargon. Memory aids such as reminders and prompts were also discussed, and participants responded positively to demonstrations/explanations of alternative modes of practice. Accessibility features such as these have been noted in other clinical populations including dementia [54] and MS [55], with authors suggesting that such modifications could be made without altering the substance, or ‘active ingredient’ of the intervention.

Fatigue was reported by most participants reflecting its known prevalence following stroke [56]. Partly due to its unpredictable nature, fatigue can make engagement with rehabilitation interventions difficult [56], as was found in this study. In a study of adults with brain injury, including stroke, researchers found that a standard MBSR course was acceptable and offered a promising non-pharmacological treatment for amelioration of mental fatigue [57]. Participants in this study suggested that shorter sessions and frequent breaks were essential to support engagement, especially in the presence of fatigue. 

Commonly after stroke, people experience difficulties with walking and mobility more generally. Participants in this study found the mindfulness walking practice physically and emotionally challenging. This was in part due to environmental considerations—in one venue there was insufficient space to allow people to practice mindful walking comfortably. However, issues with confidence, balance, altered sensation, and/or motor function also impacted participants’ ability and/or willingness to engage with this practice. Similarly, Merriman and colleagues [58] found that stroke survivors with hemiplegia found physical practices challenging where the instruction was not sensitive to post-stroke sensory and motor experiences. A finding echoed in a qualitative review of MS studies [55] in which participants described a need for trainers to *‘take on board disability’,* suggesting that careful framing, offering alternatives, and adapting movement practices according to ability, comfort or preference. Helping participants to work within their limitations and not push beyond their capabilities can enhance accessibility for people with physical limitations [22]. For example, offering the option of seated mindful movement and adapting the preparation for, and the conduct of, physical practices can help people with cognitive and/or physical limitations better engage with a practice [59,60]. In this study, amendments addressed issues of preparation, pace of delivery, guidance regarding alternative approaches, including visualization, stroke-aware delivery, and uplifting instruction, which participants found helped them better engage with the practice.

The authorized MBSR curriculum opens with an introductory session [22] during which participants are familiarized with what MBSR is and what it is not, advised about the structure of the course, and what to expect in terms of content. They also meet the trainer and other group members. In this study the introductory session was adapted to help ease pre-course anxieties and enhance accessibility. This was also an important aspect to consider in the context of the Train the Trainer package which was being developed alongside the adapted intervention. Drawing on the experience and advice of the expert panel and an understanding that telling stories is important and allows people to explain who they are and what experiences have shaped them and brought them to where they are [25], it was determined that giving people space to tell their stroke ‘story’ was an essential precursor to ‘freeing them up’ to fully engage with the course material. This finding was reflected in development work conducted by Jani and colleagues [27]. Consequently, the orientation, or introductory session, was adapted to invite the telling of participants stories in an open, non-judgmental and supportive environment. Changes were made to the Introductory pre-course pack for participants to accommodate this shift in focus of the Introductory session. In addition to practical information about transport options, the venue, accessible toilets, and so on, a brief information sheet *What is mindfulness?* was further adapted, using Plain English reinforced by illustrative graphics and other accessibility features.

Ensuring the accessibility of project and course information and other supporting materials was of paramount importance in this development work. Stroke survivors can experience various communication impairments as a consequence of stroke or due to other co-morbidities and disorders [61]. There can also be additional problems with literacy as suggested by the low rates of adult literacy across the UK, estimated as being between 12% (Wales, in 2010) and 26.7% (Scotland, in 2009) [62]. Consequently, enhancing the accessibility of an intervention such as MBSR is not only important for people affected by stroke, but has applicability and relevance across populations and health conditions, especially conditions which can affect cognitive function and communication, e.g., Multiple Sclerosis, Dementia, Long COVID.

Prior to undertaking this development work it was anticipated that adopting a dyadic approach to delivery, i.e., stroke survivors and family members attending together, would support stroke survivor participation because a co-participating ‘other’ could provide reminders and encouragement, discuss the practices and other aspects of course content, or provide practical assistance—with travel, for example. A feasibility study exploring Mindfulness Based Cognitive Therapy found involvement of a partner was perceived as potentially beneficial for the partner themselves, as well as for promoting adherence to home practice in the stroke survivor [58]. Contrary to expectation, in this adaptation study many participants preferred not to identify an ‘other’ to accompany them; however, when this topic was explored further it transpired that they did perceive peer support from within the group as being important. A recent mixed methods review [63] found contradictory evidence about the value of attending MBIs in a dyad. Quantitative data showed mixed results on interpersonal factors, whilst the qualitative synthesis found enhanced engagement with MBI and improved interpersonal communication. Similarly, a recent stroke rehabilitation intervention development study found conflicting evidence regarding the acceptability and utility of dyads as a mechanism to support adherence and behaviour change [17], echoing findings from an exploratory stroke secondary prevention study [64] in which family members were found to variously exert a positive or a negative influence on the behaviour of the stroke survivor participant.

In this adaptation work we tested a community and social media-based recruitment strategy to recruit stroke survivors and family members to the study. The resultant participant sample, whilst not representative of Western stroke clinical populations in terms of socio-economic profile and ethnic diversity, is reflective of stroke mind–body research, e.g., yoga and mindfulness. For example, a review of MBIs and stroke found participant age ranged from 52.5 (median, 38–65) to 64.8 years (mean) [26], and Jani and colleagues [27] reported participants’ mean age as 56.3 years (SD = 10.9). In the UK, the challenges associated with recruiting representative stroke populations are well documented and potential solutions have been mooted, which will be explored in future work.

Although we were engaged in early-stage development work, we were interested to ascertain whether participants considered a stroke-specific adaptation of an MBSR course to be feasible (in the context of their individual, personal circumstances), appropriate, and meaningful. The qualitative data suggest that most participants did find it feasible, appropriate and meaningful; this will be further tested and explored in subsequent mixed methods feasibility research. Although effectiveness is not addressed in developmental research such as this, and it was not appropriate to measure before-and-after symptoms of mood disorder, it was of interest to note an indication that many participants (most of whom had reported being anxious and/or depressed) found the practices that they had ‘tasted’ had either helped them manage symptoms of anxiety and depression a little better, for example in giving them mind ‘stilling’ or calming tactics to use in their busy day-to-day lives, or that the practices held the promise of being therapeutically beneficial in the longer-term if the individual pursued some form of mindfulness practice in future. 

### 4.1. Limitations

At the time of the study, MBSR courses were typically delivered by two trainers working together. As we had only one trainer working with each group, ML (first author) accompanied the trainers to provide participants with practical support should they have needed (e.g., mobility). ML’s presence in the group could have influenced subsequent data collection; however, ML was not involved in the conduct of focus groups, which were organised and moderated by BD and LDA. The study sample may be considered unrepresentative of a general stroke population; however, the intended target audience was community-dwelling stroke survivors who were experiencing symptoms of anxiety and depression as this group had been identified as an underserved population with an expressed unmet need for psychological support interventions. We believe we achieved that goal but the potential for HEADS: UP to target a more broadly inclusive participant profile will be explored in future work, as will models of delivery. 

Analysis of data from this development study informed subsequent optimisation adaptations to research processes and to the course. At the end of this study adaptations made in the context of this development work were extrapolated and applied across a full MBSR course to create the 9-week HEADS: UP course. As this was a development study, we were unable to explore accessibility and appropriateness of the fully adapted course; however, applying key principles across material is a method which has proved effective in co-design studies with stroke survivors [40,41]. The feasibility, accessibility, and appropriateness of HEADS: UP has now been tested in two non-randomized studies (face-to-face; online) and will be reported elsewhere.

### 4.2. Implications for Further Research

Inconsistent findings across stroke rehabilitation research studies suggest that further research exploring the dyad as a mechanism of behaviour change in stroke rehabilitation interventions is warranted. 

The amendments and accessibility features incorporated into the HEADS: UP course and supporting materials have been extrapolated from samples of content and materials—testing of these in a non-randomized feasibility study is required before they can be used in a large-scale RCT.

### 4.3. Implications for Practice

Interventions to equip stroke survivors and their families with skills to help them self-manage symptoms of anxiety and depression may offer accessible and appropriate options for community-dwelling adults and should be considered for use in community care.

Stroke awareness and accessibility training is essential for practitioners delivering such interventions.

## 5. Conclusions

Mood disorder is a common consequence of stroke with wide reaching impact on individuals, families, and communities; however, widespread provision of support services is lacking. HEADS: UP, a 9-week MBSR course co-developed with people affected by stroke, may provide a feasible, appropriate, and meaningful self-management intervention to help alleviate limiting effects of mood disorder. 

## Figures and Tables

**Figure 1 healthcare-11-00355-f001:**
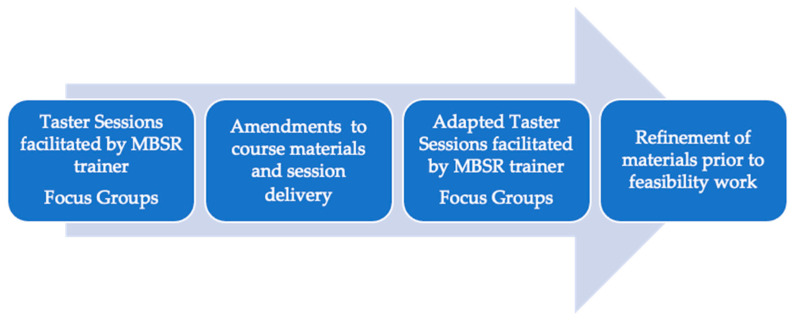
Iterative processes: HEADS: UP development study.

**Table 1 healthcare-11-00355-t001:** Stroke survivor demographic details.

	Stroke Survivors *n* = 25
**Gender**	
Female	16 (64%)
Male	9 (36%)
**Age** (*n* = 24)	Median: 62 years
	IQR 55.8 to 70.8
	Range 51–75
**Education**	
Secondary school	6 (24%)
College	12 (48%)
Undergraduate degree	5 (20%)
Masters degree	2 (8%)
**Employment**	
Full-time	3 (12%)
Part-time	2 (8%)
Not working	8 (32%)
Retired	12 (48%)
**Time after stroke**	Median: 46 months
	IQR 12.5–129.5
	Range 2–276
**Living arrangements**	
Alone	4 (16%)
With family	21 (84%)
**Anxiety** (‘yes’ in response to a yes/no question)	19 (76%)
**Depression** (‘yes’ in response to a yes/no question)	15 (60%)

**Table 2 healthcare-11-00355-t002:** Family member demographic details.

	Family Members (*n* = 7)
**Gender**	
Female	5 (71.4%)
Male	2 (28.6%)
**Age** (*n* = 24)	Median: 61.5
	IQR 50.3 to 79.8
	Range 33–91
**Education**	
Secondary school	1 (14.3%)
College	3 (42.9%)
Undergraduate degree	2 (28.6%)
Masters degree	1 (14.3%)
**Employment**	
Full-time	2 (28.6%)
Part-time	1 (14.3%)
Not working	1 (14.3%)
Retired	3 (42.9%)
**Living arrangements**	
Alone	1 (14.3%)
With family	6 (85.7%)
**Anxiety** (‘yes’ in response to a yes/no question)	5 (71.4%)
**Depression** (‘yes’ in response to a yes/no question)	4 (57.1%)

**Table 3 healthcare-11-00355-t003:** Illustrative Excerpt from the Intervention Modelling Matrix.

Core Course Elements	Qualitative Data: October	Proposed Amendments	Qualitative Data: April	Amendments Made, Tested, and Retained
Mindful movement	Just thinking about moving your legs and arms and you’re not doing it, it could be another way of, I don’t know, the brain-limb thing, a way of doing exercise but you’re not tiring yourself out so much (FM12)	TtT training to emphasize offering alternatives ways of engaging with practice(s) including visualization Adapt manual text to reflect alternative approaches and methods	But [the Trainer] tried various techniques, visual techniques. I’m paralyzed on my left side, so I’m limited with my movement, but I can do visual … it was quite good, the adaptation (SS19) Just sitting [seated movement] and your hands are down there and then you start to rotate your hands and then rotate them in different directions … he definitely gave us different ideas on how to do things (SS38)	TtT training emphasizes offering alternatives ways of engaging with practice(s) including visualization (see illustrative text below) Manual text reflects different ways of approaching the practice, including visualization, e.g., If you feel you are unable, or do not want to try a movement, it may be helpful to imagine the movement. Explore the movement without physically moving. Make sure your feet are placed flat on the floor, or a cushion. If you use a wheelchair, think of the footplates as the floor.
Walking meditations	Walking, it’s a lot of effort, you know … for me it’s a big effort because you’ve got to concentrate, your hips, your backside … so I can’t do two different things (SS8)	TtT training to emphasize offering alternatives ways of engaging with practice(s) including visualization Adapt manual text to reflect alternative approaches and methods	What I found with it was everybody concentrating much better … and I never heard the ticking clock [it had been] quite a distraction, but I never heard it at all because I was concentrating so much on the walking … I found it very helpful (SS2)	TtT training emphasizes need to offer alternative ways of engaging with practice(s) including visualization (see examples above) Manual text reflects different ways of approaching the practice, including visualization (see examples above)
Incorporating mindfulness into daily life	I would, liked to have got more information on was just how do you actually, you know, what do you do on a day-to-day basis? Are you meant to take ten minutes in the morning and do that …? How does it sort of practically … merge into your life? (FM36) I would probably need to write myself a note, that I could see, to remind me, ‘oh I’m supposed to be doing that’ … because I’m finding … unless you remind me about it, that day [I forget] (SS28)	Ensure ‘hints and tips’ sections are interspersed throughout the manual. Use emboldened text to emphasise examples of how to incorporate practice into daily life/how to start developing a mindfulness ‘habit’ mentions. Provide participants with colourful dots ahead of session 3 Encourage participants to work with an ‘other’, e.g., a fellow student, a family member, friend TtT training to emphasize reminding participants how they can fit mindfulness practice into everyday life, and illustrate with a ‘live’ example	[The trainer] definitely gave us different ideas on how to do things, and one of the things he was on about [was] … he can do it [mindfulness] just sitting waiting for a train! If you get two, three, four minutes … he can drift away, like just now [when we were coming into the room, taking our coats off], and that’s something you’d never think about (SS4)	Hints and tips sections incorporated, highlighted by being in a colored box with a colored outline Emboldened text referring to daily practice, e.g., Our breath is always with us. Researcher Session Delivery pro forma amended to include reminder to send out colourful dots ahead of session 3 Revised text encourages participants to consider working with an ‘other’ to help them remember to practice, e.g., Let us think about the support you have and the support you may need to set in place. This could be support from your buddy or your family and friends. Incorporated into TtT training–trainer to model engaging in a mindfulness practice whilst everyone is settling in, chatting, taking coats off, etc., followed by revelation and discussion during the session
Course materials	… I think we’re all the same. And that is … ‘What was that about? What was that…?’ and sometimes, my memory is very bad. So, you have to be able to go and reference something, either online or … on paper. So, you can’t always memorize the way you used to, so it’s good to have something [information sheet or manual] that you can [say], ‘Okay, that’s it there’ (SS1) Although I can read the written word, some days, that’s quite hard, and there’s sometimes I can’t remember it, so I’m much better with a video. So sometimes, online, like a lot of the information I looked at to do with stroke, I learned because I saw videos online (SS11)	Provide session material in advance of each session Develop post-session email templates to remind participants of session content, personal practice suggestions for the coming week, time and date of next session, how to contact trainer Provide suggested resources: e.g., Mindfulness Scotland website Develop videos to support delivery	… it was very welcoming, having that [introduction to mindfulness sheet]. And it gave us more information, without the jargon, without lots of jargon, yeah. (SS7) What I found most helpful out of all the information you sent was the links to the websites, so I used the Mindfulness in Scotland one (SS11)	Researcher Session Delivery pro forma schedules provision of each session’s material in advance of each session Developed post-session email templates as per column 2 Provide list of recommended resources: e.g., Mindfulness Scotland website Insufficient resources available
**General accessibility considerations to be applied across course materials** Accessibility Include some Top Tips, e.g., set aside chunks of time to read and absorb the material; don’t try to do it all at once. Think about using symbols throughout, as in Selfhelp4stroke.org, e.g., standard information symbol for ‘more information’; a symbol for ‘something to think about’, ‘hints and tips’, ‘reminders’—but not too many/use sparingly. References: Scottish Accessible Information Forum, 2014; Plain English Campaign, 2009				
Aphasia-friendly Use emboldened text to draw attention to key words and concepts. Use images to illustrate meaning and support understanding. References: Stroke Association, 2012; Haw, 2017				
Behaviour change Use repeated affirmations of progress, e.g., Well done! and prompts to ‘self-praise’, e.g., Congratulate yourself for spending time … (Feedback and Monitoring; Associations). Provide optional Support Tools and encourage regular completion/review, e.g., a checklist for daily practice(s) (Goals and Planning; Feedback and Monitoring). References: Michie et al. 2013				

## Data Availability

The data presented in this study are available on request from the corresponding author. The whole data set is not publicly available as this would violate the agreement to which the participants consented. Participants consented to use of anonymized verbatim quotes in dissemination work, and to destruction of the study data five years after completion of the study.

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
