# Peer review of "The HEADS: UP Development Study: Working with Key Stakeholders to Adapt a Mindfulness-Based Stress Reduction Course for People with Anxiety and Depression after Stroke"

_healthcare, 2023, doi:10.3390/healthcare11030355_

Round 1

Reviewer 1 Report

The manuscript reports the adaptation process of a standard MBSR course for people with depression and/or anxiety following stroke. Stroke survivors not receiving other rehabilitation input (n28) and their family members (n7) attended 1-2 MBSR taster sessions. Following this, they participated in focus groups with data analysed using framework approach mapping onto TIDieRh domains + 3 newly emergent themes. 

Overall, the manuscript is clear and well written addressing an under researched topic of adapting psychological therapies through promoting ways to improve accessibility, engagement and relevance following stroke.

The Introduction provides a comprehensive overview of relevant literature and captures not only the impact on the individual but also on families and the healthcare system.

The design of the programme is carefully considering and systematic, drawing on the existing evidence base, the MRC complex intervention framework and relevant psychological theories. A few minor adaptations would further improve the manuscript:

Inclusion criteria: “Participants were required to have self-reported and/or clinically diagnosed long-term mood disorders”. What constituted long-term and how participants were asked if they had a mood disorder? Were there any restrictions on the time after stroke? Was stroke self-reported? Where there exclusions around type eg were subarachnoid haemorrhage/TIAs? Did current engagement with rehabilitation services  involve both psychological and physical rehabilitation input?

The study methodology is generally clearly described with consideration given to methodological quality such as focus groups being run by researchers not involved in intervention delivery and double coding of qualitative data.

With regard to consultation with the team of experts, how were they recruited, how many were involved and from which disciplines? How long was the recruitment period before the October and April taster sessions?

Results 

In addition to numbers consented, it would also be helpful to report the numbers that enquired about the study or were approached if available.

Qualitative themes 

I found the name of the first theme slightly confusing as the first paragraph describes anger which does not appear to be at least fully (if at all) due to stroke hence this theme appears to focus on the impact of mindfulness on psychological wellbeing and the consequences of stroke. It is also not entirely clear how/if the results mapping onto the TIDieR domains were reported or if there were incorporated into the newly emergent themes.

Discussion

Start of second paragraph, please include physical impairment in the first sentence to summarise the main themes for completeness as mobility is not discussed until paragraph four.

Implications for practice and research have been considered to some extent. Because the participants in this development study were recruited from the community without any formal assessment of mood disorder, there should be some discussion of the proposed scope in relation to future intended target group and delivery mode (clinical services vs 3rd sector, level of clinical mood disorder) as well as discussion around limitations that it is possible that could have been a group with mild symptoms (approx 40% of those non-retired working). In relation to research implications, given that fatigue was identified as a theme, is this something you feel would be helpful to measure as part future MBSR studies in stroke?

This manuscript would be a helpful addition to accompany findings from the two RCTs referenced in the discussion given that reiterative development of interventions is an important step within the MRC complex intervention framework yet often reported in little detail in the literature or lacking involvement from those affected by the condition being investigated. 

Author Response

Reviewer 1 comments and authors’ responses

The manuscript reports the adaptation process of a standard MBSR course for people with depression and/or anxiety following stroke. Stroke survivors not receiving other rehabilitation input (n28) and their family members (n7) attended 1-2 MBSR taster sessions. Following this, they participated in focus groups with data analysed using framework approach mapping onto TIDieRh domains + 3 newly emergent themes. 

Overall, the manuscript is clear and well written addressing an under researched topic of adapting psychological therapies through promoting ways to improve accessibility, engagement and relevance following stroke.

The Introduction provides a comprehensive overview of relevant literature and captures not only the impact on the individual but also on families and the healthcare system.

The design of the programme is carefully considering and systematic, drawing on the existing evidence base, the MRC complex intervention framework and relevant psychological theories.

AUTHORS’ RESPONSE: Thank you for your comments

A few minor adaptations would further improve the manuscript:

Inclusion criteria: “Participants were required to have self-reported and/or clinically diagnosed long-term mood disorders”. What constituted long-term and how participants were asked if they had a mood disorder? Were there any restrictions on the time after stroke? Was stroke self-reported? Where there exclusions around type eg were subarachnoid haemorrhage/TIAs? Did current engagement with rehabilitation services involve both psychological and physical rehabilitation input?

AUTHORS’ RESPONSE:

‘long-term’ is an error and has been removed from line 166

Time post-stroke was not restricted, and type of stroke was not an inclusion criterion - the only requirement was for self-report. The text has been amended as follows: Participants were required to have self-reported and/or clinically diagnosed stroke and self-reported and/or clinically diagnosed mood disorder (line 166)

We required participants to not be accessing any rehabilitation services. The text has been amended to read: We …  excluded potential participants if they … were currently accessing any rehabilitation service (line 170)

The study methodology is generally clearly described with consideration given to methodological quality such as focus groups being run by researchers not involved in intervention delivery and double coding of qualitative data 

With regard to consultation with the team of experts, how were they recruited, how many were involved and from which disciplines? How long was the recruitment period before the October and April taster sessions?

AUTHORS RESPONSE:

Sentences explaining the composition of the team of experts and the route to recruitment has been inserted (line 236-239)

The team of experts comprised a diverse range of stakeholders (i.e. lay representatives, representatives of third sector organisations, clinicians, and MBSR trainers). The individual experts were network contacts or were recommended by a network contact.

Participants were recruited and screened over a period of 6 weeks in August and September 2017. A sentence to this effect has been added at line 253: Recruitment occurred over 6 weeks in August and September 2017. 

Results 

In addition to numbers consented, it would also be helpful to report the numbers that enquired about the study or were approached if available.

AUTHORS’ RESPONSE:

We received 38 expressions of interest. The text has been amended (line 298-299) to include this detail: We received 38 expressions of interest. Three were lost to contact; 35 (92.1%) gave written informed consent to participate in the study.

Qualitative themes 

I found the name of the first theme slightly confusing as the first paragraph describes anger which does not appear to be at least fully (if at all) due to stroke hence this theme appears to focus on the impact of mindfulness on psychological wellbeing and the consequences of stroke. It is also not entirely clear how/if the results mapping onto the TIDieR domains were reported or if there were incorporated into the newly emergent themes.

AUTHORS’ RESPONSE:

We agree, the theme label could be improved on! Therefore, we have amended the text (in lines 279 and 285) to read: The role of Mindfulness in mitigating stroke risk factors and effects

We have amended the text slightly at line 265 to make clear that only the inductive themes are reported in this paper: The inductive themes only are reported here, supported by verbatim quotes.

Discussion

Start of second paragraph, please include physical impairment in the first sentence to summarise the main themes for completeness as mobility is not discussed until paragraph four.

AUTHORS’ RESPONSE:

The suggested change has been made at line 464: In addition to the physical effects of stroke, stroke survivors described …

Implications for practice and research have been considered to some extent. Because the participants in this development study were recruited from the community without any formal assessment of mood disorder, there should be some discussion of the proposed scope in relation to future intended target group and delivery mode (clinical services vs 3rd sector, level of clinical mood disorder) as well as discussion around limitations that it is possible that could have been a group with mild symptoms (approx 40% of those non-retired working).

AUTHORS’ RESPONSE: Thank you for your comments. we have addressed some if these by inserting the following text at lines 580-586: The study sample may be considered unrepresentative of a general stroke population however the intended target audience was community-dwelling stroke survivors who were experiencing symptoms of anxiety and depression as this group had been identified as an underserved population with an expressed unmet need for psychological support interventions. We believe we achieved that goal but the potential for HEADS: UP to target a more broadly inclusive participant profile will be explored in future work, as will models of delivery.

In this paper we have reported on presence of symptoms rather than symptom severity so have not embarked on that discussion here. In future work we will assess severity of mood disorder and work to determine which stroke survivors would benefit most from HEADS: UP.

In relation to research implications, given that fatigue was identified as a theme, is this something you feel would be helpful to measure as part future MBSR studies in stroke?

AUTHORS’ RESPONSE: We are grateful for this suggestion, and it is something we have debated over the years. We are currently planning a definitive trial of HEADS: UP and will make decision regarding this issue in the coming months.

This manuscript would be a helpful addition to accompany findings from the two RCTs referenced in the discussion given that reiterative development of interventions is an important step within the MRC complex intervention framework yet often reported in little detail in the literature or lacking involvement from those affected by the condition being investigated.

AUTHORS’ RESPONSE: Thank you – we agree!

Reviewer 2 Report

Overview: The article focuses on the structured mindfulness-based stress reduction (MBSR) courses. The research focused on stroke survivors and family members with symptoms of anxiety and/or depression to take part in a co-development study comprising two rounds of MBSR ‘taster’ sessions, followed by focus groups in which views were sought on the practices sampled. This is a very pertinent issue and the article deals well with this. This article could be a starting point for further research.

Theoretical frame: It offers a robust theoretical frame and a well-organized and convincing discussion of the findings. This is a very interesting article where policy implications are well explained. A clear stating and focusing of the argument is provided.

Methods: Methods are appropriate and the fit between theoretical discussion and methodology is well formulated. 

Results: Results are linked suitably to the other sections of the article. A well-organized and compelling discussion section is provided as well.

Language: The authors have paid attention to the clarity of expression and readability, such as sentence structure. The quality of communication is good.

Author Response

Reviewer 2 comments and authors’ responses

Overview: The article focuses on the structured mindfulness-based stress reduction (MBSR) courses. The research focused on stroke survivors and family members with symptoms of anxiety and/or depression to take part in a co-development study comprising two rounds of MBSR ‘taster’ sessions, followed by focus groups in which views were sought on the practices sampled. This is a very pertinent issue and the article deals well with this. This article could be a starting point for further research.

Theoretical frame: It offers a robust theoretical frame and a well-organized and convincing discussion of the findings. This is a very interesting article where policy implications are well explained. A clear stating and focusing of the argument is provided.

Methods: Methods are appropriate and the fit between theoretical discussion and methodology is well formulated. 

Results: Results are linked suitably to the other sections of the article. A well-organized and compelling discussion section is provided as well.

Language: The authors have paid attention to the clarity of expression and readability, such as sentence structure. The quality of communication is good.

AUTHORS’ RESPONSE: We are very grateful for your comments - thank you.

Reviewer 3 Report

This is an interesting qualitative study collecting information on two-time frames (October 2017 and April 2018). I have the following comments:

1. Abstract,

1.1. Include the data collection period.

1.2. You mentioned the total number of informants involved in the study, but not all participated in April and October interviews. Better mentioned here.

2. Materials and Methods

2.1. In the focus group discussions, "Moderated by BD and LdA" and audio recorded, what is the role of the two moderators, or are they separated into two groups, or is one observing and the other initiating the discussion? Suggest elaborating more on this.

2.2. In the Iterative process section, you wrote BD and ML are independently immersed in the data by ..." So, ML is not involved in the interview, and why is not LdA instead ML? I have no issue with BD doing this, but why does ML do it instead of LdA? Is this related to the rigor or reliability of your methodology?

3. Results

3.1. What is the assessment tool to determine informants' anxiety and depression in Tables 1 and 2? Perhaps need to put it in the footnote?

3.2. Will informants who are anxiety and depressed their responses are different from those without any anxiety and depression during the two periods of time? Perhaps can elaborate more on this in the results and discussion section if this occurs.

3.3. In Table 3, the 2nd column is April 2018, and the 4th column is Oct 2017? Is it Oct 2017 first and then April 2018?

Author Response

Reviewer 3 comments and authors’ responses

This is an interesting qualitative study collecting information on two-time frames (October 2017 and April 2018). I have the following comments:

  1. Abstract,

1.1. Include the data collection period.

AUTHORS’ RESPONSE: The following text has been inserted at lines 24-25: Data were collected in October 2017 and May 2018 and were …

1.2. You mentioned the total number of informants involved in the study, but not all participated in April and October interviews. Better mentioned here.

AUTHORS’ RESPONSE: We have inserted text to that effect in line 29: and three (10%) were unable to attend either round.

  1. Materials and Methods

2.1. In the focus group discussions, "Moderated by BD and LdA" and audio recorded, what is the role of the two moderators, or are they separated into two groups, or is one observing and the other initiating the discussion? Suggest elaborating more on this.

AUTHORS’ RESPONSE: Further elaboration has been provided, as follows, at lines 195-200: Focus group discussions, moderated by BD and co-moderated by LdA (second and third authors) were audio recorded and then transcribed. The role of the moderator (BD) was to ‘manage’ the focus group e.g. ensure all participants had the opportunity to contribute to the discussions. The co-moderator (LdA) supported BD e.g. by making field notes and asking any supplementary questions, if appropriate; neither were involved in the taster sessions.

2.2. In the Iterative process section, you wrote BD and ML are independently immersed in the data by ..." So, ML is not involved in the interview, and why is not LdA instead ML? I have no issue with BD doing this, but why does ML do it instead of LdA? Is this related to the rigor or reliability of your methodology?

AUTHORS’ RESPONSE: To enhance rigour ML conducted the analysis with BD. We have amended the text at line 210 as follows: In the interests of rigor, two researchers (BD and ML) independently immersed themselves …

  1. Results

3.1. What is the assessment tool to determine informants' anxiety and depression in Tables 1 and 2? Perhaps need to put it in the footnote?

AUTHORS’ RESPONSE: The following text has been inserted at line 182: … symptoms of anxiety (yes/no), symptoms of depression (yes/no) … and into Tables 1 and 2: (‘yes’ in response to a yes/no question)

3.2. Will informants who are anxiety and depressed their responses are different from those without any anxiety and depression during the two periods of time? Perhaps can elaborate more on this in the results and discussion section if this occurs.

AUTHORS’ RESPONSE: This data was not recorded so we are not able to report it in this paper.

3.3. In Table 3, the 2nd column is April 2018, and the 4th column is Oct 2017? Is it Oct 2017 first and then April 2018?

AUTHORS’ RESPONSE: It is indeed! Thank you for noticing this - the columns are now correctly labelled!